# Analysis and Comparison of Bio-Oils Obtained by Hydrothermal Liquefaction of Organic Waste

**Yuliya Kulikova** [1,*] , **Marina Krasnovskikh** [2] , **Natalia Sliusar** [3] , **Nikolay Orlov** [1] and **Olga Babich** [1,*]

1. Institute of Living Systems, Immanuel Kant BFU, 236016 Kaliningrad, Russia
2. Department of Inorganic Chemistry, Chemical Technology and Technosphere Safety of Perm State National Research University, 614990 Perm, Russia
3. Environmental Protection Department, Perm National Research Polytechnic University, 614000 Perm, Russia
* Correspondence: kulikova.pnipu@gmail.com (Y.K.); olich.43@mail.ru (O.B.); Tel.: +7-912-7849-858 (Y.K.)

**Abstract:** This paper presents an analysis of bio-oil quality depending on the type of input biomass, the process conditions and the catalytic systems used. Analysis of various catalytic system choices showed the prospects of using nickel and iron metal salts as homogeneous catalysts given that their use provided increases of 24.5% and 22.2%, respectively, in the yield of light-boiling bio-oil fractions (with a boiling point of up to 350 °C). Composition analysis of the bio-oils carried out using gas chromatography and nuclear magnetic resonance spectroscopy showed that fatty acids are the predominant group of substances in bio-oils produced from sewage sludge. Bio-oil synthesized from bark and wood waste contains phenolic alcohols and a limited range of cyclic hydrocarbons as the main components. In bio-oil produced from macroalgae, oxygen and nitrogen compounds of the piperazinedione and amides type are predominant. The sulfur and nitrogen content in all types of bio-oils is at an acceptable level. The results allow researchers to assert that organic waste processing enables production of sufficiently high-quality fuel, which can then be jointly processed with natural oil. Bio-oil produced from secondary sludge has the best quality, characterized by a high content of low-weight aliphatic compounds (with a boiling point of up to 350 °C), along with insignificant levels of nitrogen, sulfur and oxygen.

**Keywords:** sewage sludge; organic waste; hydrothermal liquefaction; catalysis; bio-oil





## 1. Introduction

Currently, humanity is faced with two divergent problems: on the one hand, the level of social and technological development requires an increasing volume of production and use of hydrocarbons, while, on the other hand, the rapid decline in fossil hydrocarbon reserves, along with the growing impact of climate change, highlights the need to search for alternative sources of hydrocarbons. This has led research on hydrothermal liquefaction (HTL) of various organic waste products to become increasingly relevant. As is well-known, the main distinctive advantage of this technology is that it offers the possibility to process wet biomass [1].

Use of HTL technology for processing various types of plant biomass and algae provides a fuel yield of 10–20% [2,3]. It should be noted that, from the point of view of solving environmental problems, it is more efficient to use organic waste as a raw material for hydrothermal conversion. Transformation of organic waste to produce fuel, while simultaneously solving technological problems of biofuel production, will also solve a number of environmental problems via reduction in greenhouse gas emissions generated by waste degradation, return to economic use of territories occupied by waste disposal sites and other such positive outcomes.

The nature of the HTL process allows it to be used for raw materials with high moisture content and materials from a wide range of sources. Numerous studies have been

published on the HTL of forestry and woody waste [4–7], agricultural waste [8–11], sewage sludge [1,12–22], manure [23,24] and algae [3,16,25,26]. Choice of feedstock depends on a variety of factors, such as availability, cost of production and composition of the feedstock.

For example, the HTL of lignocellulosic materials leads to formation of a complex mixture of chemicals containing a significant proportion of oxygen [22]. After conversion of sewage sludge, the fuel contains a significant amount of nitrogen [21]. Algae contain carbohydrates, lipids and proteins [26–28] that break down into various organic chemicals, some of which contain nitrogen caused by deamination of amino acids from proteins. The fuel yield also undoubtedly varies depending on the type of biomass.

Ensuring maximum yield and calorific value of fuel is a central problem to address when it comes to widespread use of hydrothermal liquefaction for processing wet biomass. In this study, we have done our best to cover the many different types of biomass and organic waste generated in the Kaliningrad region and suitable for hydrothermal liquefaction. However, we chose to highlight bark and wood wastes with a significant storage period (20–50 years), as well as primary and secondary sludge, as our main areas of focus. Other types of biomasses were regarded as complementary in construction of the process for converting the aforementioned main types of waste.

If we consider various approaches to processing these wastes, we will see that a number of papers [18,19] have been devoted to study of converting wastewater sludge via HTL, confirming the possibility of a fuel yield of 30% without use of catalysts [20]. Use of alkaline and metal-containing catalysts makes it possible to increase the fuel yield up to 50% [21,22]. According to numerous studies, bio-oil from wastewater treatment sludge has comparatively good quality, with a low content of nitrogen (3–7%) and oxygen (10–17%), and a higher heating value 22–48 MJ/kg. [4,25,29]. However, the detailed composition of bio-oil from sludge, namely the main groups of substances contained and the content of various fractions by boiling points, is not described in the literature.

Feng et al. [30] studied the HTL of bark from various species in water–ethanol solvents. It was shown in their work that the main components of the bio-oil were phenolic compounds, and the HHV was 25–39 MJ/kg. Sandquist et al. [31] studied the HTL of bark-containing wood waste from pine and spruce at 350 °C and 400 °C. It was found that the maximum achievable bio-oil yield was 25–30% d.m. Questions related to hydrothermal liquefaction of old bark and wood waste (BWW) are not discussed in the literature. Therefore, it is especially important to carry out further research to assess the yield and composition of bio-oil after processing of old BWW. Previously, we conducted a set of studies on composition and properties of bark and wood waste (BWW) [4]. The outcome of our work proved the possibility of using BWW for fuel production, but the HTL results summarized here have not been previously presented.

This paper lays out our conclusions from additional comprehensive studies examining biofuel composition after the process of catalytic and non-catalytic conversion of sewage sludge, BWW, aquatic plants, soy husk and algae. The aim of this study is to evaluate and compare the prospects of using various organic wastes as raw materials for HTL conversion and to analyze their potential for high-quality fuel production.

## 2. Materials and Methods

### 2.1. Materials

The following types of biomass were selected as raw materials for the hydrothermal liquefaction process:

- microalgae (*Chlorella vulgaris*),
- macroalgae,
- aquatic plants of three species (*Schoenoplectus lac.*, *Typha angustifolia* L., *Phragmites vulgaris*);
- soy production waste (soy husk)
- BWW from dumpsites
- sludge from primary clarifiers
- sludge from secondary clarifiers

The choice of biomass for processing was primarily due to the need to solve environmental problems in the Kaliningrad region. Factors considered when selecting types of biomass included the volume of waste, as well as the possibility of cascade processing and obtaining additional value-added products. Aquatic plants were chosen because the issue of eutrophication in Baltic Sea lagoons is acute in the region, and timely removal of aquatic vegetation will partially reduce the biogenic load.

Microalgae were grown in the laboratory in a nutrient aqueous medium from a standard culture of *Chlorella vulgaris*.

Macroalgae were collected from the Baltic Sea coast (from beach-cast) between July and September 2021. The predominant species of algae were *Ulva* sp. and *Polysiphonia* sp.

Samples of aquatic plants (*Schoenoplectus lacustris*, *Typha angustifolia* L., *Phragmites vulgaris*) were collected between June and August 2022 from the coast of the Curonian Lagoon in the Kaliningrad region of Russia (55°09′27.4″ N 20°51′04.4″ E). The waste was a mixture of bark and chips of coniferous species (pine and spruce) with minor destruction as a result of natural rotting processes.

Samples of soybean oil production waste (soy husk) were provided by the Sodruzhestvo Group in the Kaliningrad region of Russia. This waste is produced as a result of direct oil extraction from soybean seeds.

Sampling of BWW was conducted in July 2021 from an old bark dumpsite in Krasnokamsk, Russia by drilling to a depth of 0–17 m at three points (58°04′16.6″ N 55°45′55.1″ E). Accumulation of BWW at the dumpsite had taken place between 1936 and 2005.

Samples of dewatered sludge were received from a biological wastewater treatment plant in the city of Kaliningrad, Russia. Sludge from primary clarifiers (primary sludge) mainly consists of organic and inorganic small particles since large particles are deposited in sand traps and gratings prior to reaching the clarifier tanks. Secondary sludge represents excess sludge removed from the system in secondary clarifiers.

Before processing, all samples were dried at 60 °C, crushed in a knife crusher and sieved to achieve a fraction size of 2–4 mm. The resulting fraction was then dried to a constant weight (for 12–18 h) in an oven at 104 °C. The drying temperature of 60 °C was chosen because some authors had demonstrated that higher temperatures lead to changes in the physicochemical composition of the biomass, in particular partial destruction of proteins and lignin [32].

### 2.2. HTL Process Parameters

The studies were conducted according to the scheme shown in Figure 1. The HTL process was carried out in a 300 mL autoclave reactor (Zhengzhou Keda Machinery and Instrument Equipment Co., Ltd., Zhengzhou, China) made of non-rusting steel and equipped with an external electric heater. The reactor also had a magnetic stirrer, a pressure gauge and a thermocouple. The average heating rate was 10 degrees/min. The sealing reactor was previously purged with nitrogen. The temperature in all experiments varied within a range of 240–280 °C. The holding time was 5–60 min, the pressure was 3.2–6.5 MPa and the raw material to water ratio was 1:5–1:20 (d.m.). The base (standard) conditions were taken at a temperature of 260 °C, a time of 20 min and a raw material to water ratio of 1:10.

Based on analysis of literature data, the following were selected as homogeneous catalysts: KOH, NaOH, NaHCO3, $H_4Fe(SO_4)_2$, $CoCl_6$, $NiSO_4$, $CuSO_4$, $ZnSO_4$, $MoO_3$, $H_7[P(Mo_2O_7)_6]$. Catalyst cations included in the composition of the listed salts and alkalis were considered as target active components. MgO, zeolite, $V_2O_5$ and $Al_2O_3$ were selected as heterogeneous catalysts. The dose for all catalysts is 2 g per 10 g of raw material (d.m.). The rationale for the choice of these catalysts is presented in our previous work [21].

At the end of the process, the reactor was cooled with water to a temperature of 24 °C. Then, gas products were collected in multi-layer foil gas sampling bags. Periodic assessment of the volume of gases showed that their quantity did not exceed 200 mL (atm pressure).

The oil fraction was extracted with dichloromethane and separated from the aqueous phase using a separation funnel. The solvent was distilled using a rotary vacuum evaporator at 40 °C and a vacuum of 0.780 bar.

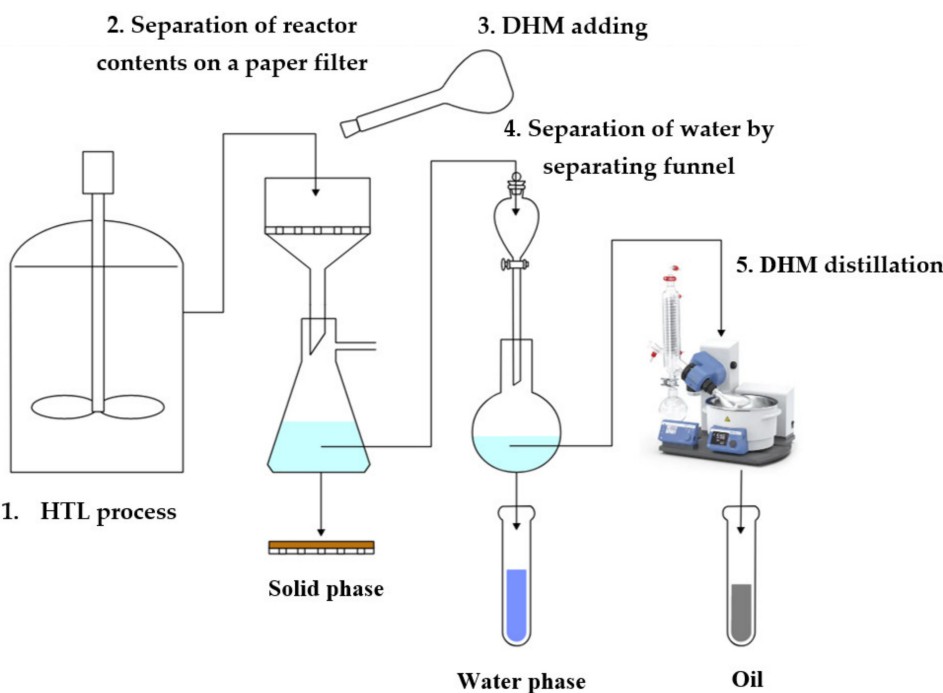

**Figure 1.** Scheme for hydrothermal liquefaction experiments.

### 2.3. Bio-Oil Composition Analysis

Elemental analysis of the bio-oils was performed using a CHNS-elemental analyzer (Elementar Analysensysteme Vario EL Cube, Elementar Analysensysteme GmbH, Langenselbold, Germany). Weighing was carried out on analytical scales with an accuracy of 0.01 mg. The content of the elements was determined based on the area of chromatographic peaks of $N_2$, $CO_2$, $H_2S$ and $SO_2$ by a calibration line constructed using standard compounds. Each sample was examined in three independent parallels. The average values are presented here. The software of the equipment manufacturer was used to process analytical data and calculate the content of elements in the samples.

Synchronous thermal analysis of bio-oils was carried out in an inert (argon) environment. It was used as an analogue of micro-distillation to clarify the fractional composition of petroleum products by boiling point. The studies were completed using a NETZSCH STA 449C Jupiter synchronous thermal analysis device. The calculations were completed in three ranges of volatile fractions with boiling points of 42–150 °C—gasoline; 150–360 °C—kerosene and diesel fuel; 360–500 °C—fuel oil.

Hydrocarbon content and bio-oil fractional composition were determined by GC/MS according to GOST ASTM D5307, D2887, D3710. Bio-oil samples were dissolved in dichloromethane and analyzed with GC/MS Agilent 7890B-5977A. Detection conditions were as follows: electronic shock, ionization energy—70 eV, gas flow rate He—1 mL/min, HP-1MS column, evaporator temperature—350 °C, input temperature—320 °C, sample volume—0.05–0.1 μL. Temperature programs were selected depending on the sample and the required separation accuracy:

- Isotherm 50 °C for 1 min, temperature gradient 50–300 °C for 125 min, isotherm 300 °C for 9 min;
- Isotherm 50 °C for 1 min, temperature gradient 50–300 °C for 50 min, isotherm 300 °C for 9 min.



The results were processed using the software supplied with the equipment. Identification of the products was carried out based on a comparison of their mass spectra with the spectra of compounds available in the NIST 2017 library attached to the equipment software.

NMR spectroscopy was performed on a Bruker Avance III HD spectrometer (400 MHz 1H, 101 MHz 13C). The choice of conditions for the solution preparation, the methodology and the procedure for processing experimental study results was based on available literature data [33–35].

Deuterated chloroform ($CDCl_3$) was used for petroleum product analysis. Chemical shifts were determined relative to hexamethyldisiloxane—1H: 0.07 ppm in $CDCl_3$, 0.28 ppm in $D_2O$ or relative to the residual signal of CDCl3 solvent 1H: 7.26 ppm, 13C: 77.16 ppm and temperature during measurement—40 °C.

Deuterium oxide ($D_2O$) (10% of the aqueous phase volume) was also used as a solvent. For aqueous solutions (10% $D_2O$ in 90% $H_2O$), the solvent signal was suppressed. The integral of the solvent area was not taken into account in the final result, which may affect the accuracy of determining the mass and molar fraction of the corresponding protons. The ranges of chemical shifts are shown in Table 1.

**Table 1.** Ranges of chemical shifts of NMR spectroscopy.

| Type of Analysis | Type of Bond | Shift (ppm) |
|---|---|---|
| NMR 1H | COOH, CHO, ArOH | 8.2–12.0 |
| | Aromatic compounds, alkynes | 6.0–8.2 |
| | Aliphatic R-OH, -$CH_2$-O, Alkenes | 4.2–6.0 |
| | Aliphatic R-$CH_2$-O, -$CH_3$-O | 3.0–4.2 |
| | Aliphatic R-$CH_2$-CH=O | 2.0–3.0 |
| | Aliphatic | 0–2.0 |
| NMR 13C | Aldehydes, ketones | 220–180 |
| | Acids and derivatives | 180–160 |
| | Aromatic compounds | 160–105 |
| | Aromatic compounds, without substitution | 140–125 |
| | Alcohols, esters, carbohydrates | 105–60 |
| | $CH_3O$ in lignin | 60–55 |
| | Aliphatic carbon | 55–1 |

## 3. Results and Discussion

### 3.1. Results of the Synthetic Oil Fractional Composition Analysis

The distribution of bio-oil components by boiling points is shown in Table 2. Estimation was carried out using thermogravimetric analysis in an inert environment as an analogue of micro-distillation. Analysis of the results allows researchers to conclude that the samples of bio-oil from sewage sludge contained 38.4–53.1% oil fraction with a boiling point of 250–350 °C (diesel-like). At the same time, the content of fractions with a boiling point of up to 250 °C (gasoline-like) was 12.0–21.7%.

The analysis of the effect of catalysts on the content of low-boiling fractions (up to 350 °C) is shown in Figure 2, which illustrates this fraction content increase in comparison with the experiment under similar conditions but without use of catalysts.

The maximum increase in the yield of low-boiling fractions was reached when nickel sulfate (24.5%) and copper (18.8%) were used as catalysts. Use of nickel cations has provided an increase in biofuel yield in various studies: Mukundan et al. showed a 50% increase in oil yield from HTL of a plant biomass and polymer mixture; in the processing of pine sawdust, Cheng et al. [36] obtained an increase in oil yield from 59% to 71% using a nickel catalyst. Thus, our results regarding catalytic conversion of biomass are fully consistent with the literature data.

Copper-based catalysts, as with all redox catalysts, undergo both oxidation and reduction reactions during the process, which lead to a change in their oxidation state. Copper

and its compounds have been repeatedly and successfully used for catalytic HTL of various biomass due to their redox catalytic core and easy regeneration, along with their high activity and selectivity. Redox catalysts are capable of producing hydrogen in situ and subsequently promote hydrogenation of reactive compounds [37]. Wang et al. achieved an increase in oil yield from 38.8% to 47.6% during HTL of sewage sludge [15].

Use of phosphorus molybdenum acid as a catalyst also provides a positive effect and a 22.2% increase in the yield of light oil fractions. Catalytic HTL of sewage sludge and *Chlorella* sp. using molybdenum-based catalysts showed a positive effect on the yield and quality of bio-oil in the study of Prestigiacomo et al. [16].

Use of alkaline catalysts leads to both a 27.5–63.4% decrease in bio-oil yield [21] and a 1–43% increase in the ratio of low-boiling fractions relative to the control. As can be seen from the results, use of alkali metal salts and bases did not have a visible effect, which is somewhat different from the results of studies presented by Basar et al. [22] and Shah et al. [14], which showed that use of ions increased the yield by 140% and 12%, respectively. In the literature, there are also opposite results showing that use of alkaline catalysts does not provide a significant effect. For instance, Malins et al. [13] only achieved a decrease in bio-oil yield from 40.4% to 38.8% when using $Na_2CO_3$. A similar result was obtained by Qian et al. [12] in their research on sodium and potassium carbonates as catalysts, which produced a negative outcome, namely a decrease in oil yield from 26.8% to 20.9–21.6%. Such unstable results are rather difficult to explain. Perhaps they are due to the fact that silt biomass differs significantly in composition.

**Table 2.** Ratio of substances with different boiling points in composition of bio-oil produced from various raw materials under different conditions.

| Type of Raw Material | Catalyst | Conditions [1] | Weight Loss in Temperature Range (%) | | | | | |
|---|---|---|---|---|---|---|---|---|
| | | | 30–170 | 170–250 | 250–350 | 350–500 | 500–800 | Above 800 |
| S. sludge | None | Biomass:water 1:5 | 3.78 | 11.78 | 41.00 | 21.90 | 4.14 | 16.74 |
| S. sludge | None | Biomass:water 1:20 | 3.41 | 12.22 | 38.35 | 21.43 | 15.13 | 9.47 |
| S. sludge | None | 10 min | 3.54 | 12.76 | 40.48 | 19.55 | 12.94 | 10.73 |
| S. sludge | None | 15 min | 2.36 | 12.68 | 43.44 | 20.27 | 13.98 | 7.27 |
| S. sludge | None | 30 min | 2.93 | 11.92 | 41.94 | 21.06 | 9.50 | 12.65 |
| S. sludge | None | 60 min | 3.55 | 12.44 | 40.03 | 18.65 | 11.81 | 13.53 |
| S. sludge | $NiSO_4$ | Standard | 3.40 | 15.60 | 51.72 | 19.70 | 2.64 | 6.94 |
| S. sludge | Zeolite | Standard | 3.00 | 13.24 | 43.22 | 21.01 | 14.67 | 4.82 |
| S. sludge | $V_2O_5$ | Standard | 2.93 | 13.16 | 50.44 | 19.85 | 12.71 | 0.91 |
| S. sludge | MgO | Standard | 4.86 | 7.17 | 19.905 | 39.05 | 6.47 | 22.54 |
| S. sludge | $CoCl_6$ | Standard | 4.15 | 13.73 | 46.98 | 21.47 | 3.24 | 10.44 |
| S. sludge | $MoO_3$ | Standard | 3.27 | 16.22 | 44.81 | 15.67 | 11.04 | 9.00 |
| S. sludge | $Al_2O_3$ | Standard | 3.60 | 12.02 | 43.46 | 20.90 | 11.52 | 8.51 |
| S. sludge | KOH | Standard | 6.63 | 15.05 | 42.25 | 18.90 | 9.33 | 7.83 |
| S. sludge | $CuSO_4$ | Standard | 3.87 | 15.26 | 48.34 | 17.53 | 11.14 | 3.86 |
| S. sludge | $ZnSO_4$ | Standard | 3.06 | 13.41 | 48.04 | 19.16 | 5.49 | 10.83 |
| S. sludge | $H_7[P(Mo_2O_7)_6]$ | Standard | 2.67 | 13.61 | 53.09 | 20.34 | 4.72 | 5.56 |
| S. sludge | $NaHCO_3$ | Standard | 4.94 | 14.22 | 37.36 | 22.08 | 8.65 | 12.75 |
| S. sludge | $NH_4Fe(SO_4)_2$ | Standard | 3.89 | 11.67 | 45.80 | 23.77 | 10.26 | 4.62 |
| P. sludge | None | Standard | 7.41 | 14.84 | 37.67 | 17.61 | 11.84 | 10.58 |
| P. sludge | $NH_4Fe(SO_4)_2$ | Standard | 9.39 | 12.71 | 31.36 | 19.53 | 13.96 | 13.04 |
| P. sludge | Zeolite | Standard | 5.28 | 12.31 | 38.64 | 23.30 | 9.99 | 10.42 |
| P. sludge | $MoO_3$ | Standard | 7.24 | 16.26 | 41.64 | 14.97 | 5.54 | 14.35 |
| P. sludge | $CoCl_6$ | Standard | 5.42 | 15.28 | 43.90 | 19.43 | 3.99 | 11.98 |
| P. sludge | $NiSO_4$ | Standard | 6.63 | 13.16 | 41.38 | 19.88 | 7.33 | 11.62 |
| P. + S. sludge (1:1) | None | Standard | 3.34 | 12.34 | 38.99 | 22.03 | 12.05 | 11.24 |
| P. + S. sludge (1:1) | $NH_4Fe(SO_4)_2$ | Standard | 3.57 | 12.76 | 39.52 | 22.56 | 11.88 | 9.71 |
| *Schoenoplectus lac.* | None | Standard | 15.24 | 16.25 | 28.05 | 15.69 | 3.11 | 21.66 |
| *Typha angustifolia* L. | None | Standard | 15.46 | 16.49 | 20.87 | 12.03 | 21.40 | 13.75 |
| *Phragmites vulgaris* | None | Standard | 14.27 | 16.55 | 25.93 | 16.75 | 9.93 | 16.58 |
| *Chlorella vulgaris* | None | Standard | 3.95 | 10.82 | 36.21 | 22.32 | 18.53 | 8.18 |
| Soy husk | None | 280 grad | 7.54 | 11.70 | 32.52 | 24.50 | 8.50 | 15.25 |

[1]—"Standard" stands for the following process conditions: 20 min, 260 °C, 4.6 MPa, biomass: water = 1:10.

Analysis of data on bio-oil yield and changes in its composition indicated that use of nickel sulfate as a catalyst provides the most promising results. This is because, along with an increase of 63.4% in the bio-oil yield relative to the control, it also provides an increase in the ratio of gasoline-like and diesel-like fractions.

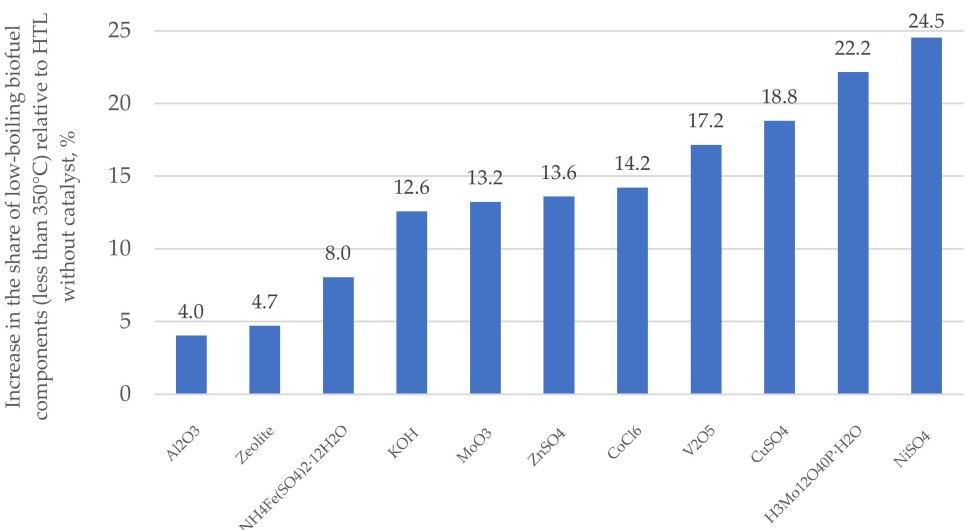

**Figure 2.** Ratio of bio-oil components with different boiling points. Raw material for HTL process: secondary sludge.

Process time and biomass volume had no reliable effect on the quality of the bio-oil produced from sewage sludge, either because the content of the individual fractions did not change or because the changes were not significant (within the margin of error).

A comparison of the fractional composition of bio-oil produced from various raw materials is shown in Figure 3. Based on the data obtained, 20 min was determined to be the optimal time for the process.

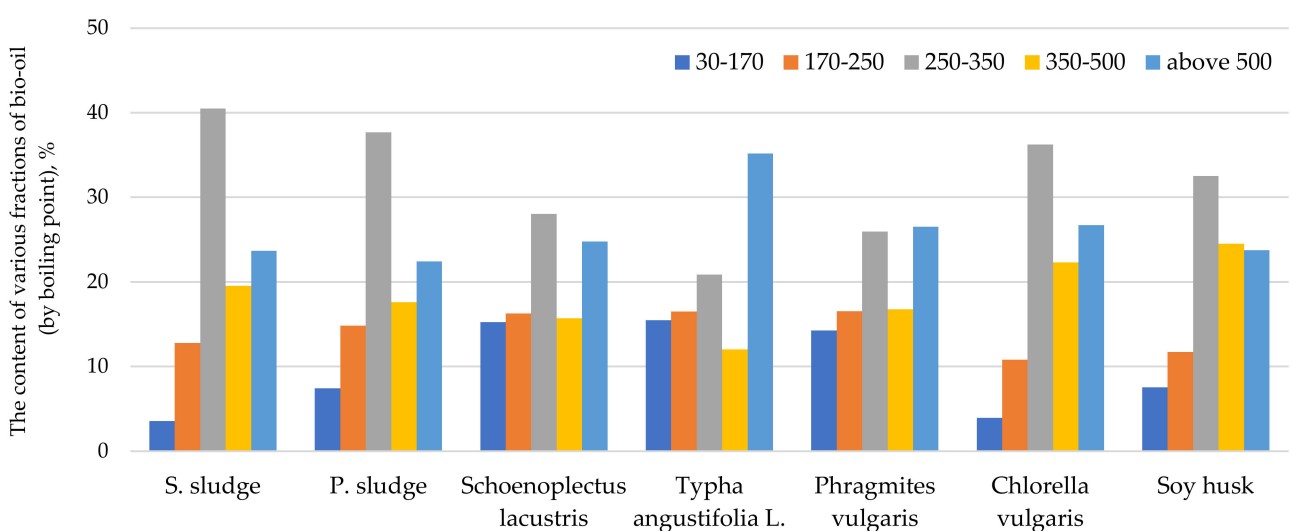

**Figure 3.** Ratio of bio-oil components with different boiling points from various raw materials.

The average content of heavy fractions in the bio-oil produced from cellulosic raw materials is much higher, which is certainly due to the high content of cellulose and lignin in comparison with primary and secondary sludge. At the same time, use of plant biomass, which contains a significant amount of soluble carbohydrates, leads to formation of volatile organic compounds (alcohols, ketones, etc.) with a boiling point of up to 170 °C. The content of volatile organic compounds in bio-oil produced from plant biomass (*Schoenoplectus*

*lacustris*, *Typha angustifolia* L., *Phragmites vulgaris*) is, on average, 4.2 times higher than in bio-oil produced from secondary sludge, which contains insignificant amounts of soluble carbohydrates. Given the lower yield of liquid bio-oil produced during plant biomass conversion (according to the results of the study, no more than 5% d.m.), the prospects for using this type of biomass are doubtful.

### 3.2. Elemental Composition of Bio-Oils

The elemental composition of the bio-oils was determined in order to analyze the content of nitrogen and sulfur as the main interfering components, as well as to assess the quality of each bio-oil. The summary results are presented in Table 3.

**Table 3.** Main chemical element content in the composition of bio-oil produced from various raw materials and using different catalysts.

| Type of Raw Material, Conditions [1] | Catalyst | Element Content, % d.m. | | | |
|---|---|---|---|---|---|
| | | C | H | N | S |
| P. sludge, 240 grad | none | $68.20 \pm 0.63$ | $9.50 \pm 0.29$ | $5.97 \pm 0.05$ | $1.06 \pm 0.07$ |
| P. sludge, Standard | $NH_4Fe(SO_4)_2 \cdot 12H_2O$ | $62.1 \pm 3.35$ | $8.63 \pm 0.27$ | $6.06 \pm 0.08$ | $1.07 \pm 0.04$ |
| P. sludge, Standard | Zeolite | $69.43 \pm 0.40$ | $9.42 \pm 0.13$ | $6.27 \pm 0.05$ | $1.04 \pm 0.02$ |
| P. sludge, Standard | $MoO_3$ | $68.85 \pm 0.36$ | $8.90 \pm 0.31$ | $6.55 \pm 0.14$ | $0.73 \pm 0.01$ |
| P. sludge, Standard | $CoCl_6$ | $70.8 \pm 1.11$ | $9.22 \pm 0.29$ | $4.93 \pm 0.10$ | $0.58 \pm 0.01$ |
| P. sludge, Standard | $NiSO_4$ | $69.62 \pm 0.15$ | $8.89 \pm 0.20$ | $6.21 \pm 0.12$ | $0.51 \pm 0.04$ |
| S. sludge, 240 grad | none | $68.34 \pm 0.94$ | $9.49 \pm 0.13$ | $2.83 \pm 0.05$ | $0.89 \pm 0.02$ |
| S. sludge, Standard | none | $63.25 \pm 2.15$ | $10.20 \pm 0.22$ | $1.77 \pm 0.15$ | $0.75 \pm 0.09$ |
| S. sludge, 10 min | none | $68.49 \pm 1.29$ | $10.11 \pm 0.09$ | $2.49 \pm 0.04$ | $0.94 \pm 0.02$ |
| S. sludge, 15 min | none | $70.2 \pm 0.06$ | $10.32 \pm 0.14$ | $2.32 \pm 0.02$ | $0.91 \pm 0.01$ |
| S. sludge, 30 min | none | $71.11 \pm 1.18$ | $10.23 \pm 0.04$ | $2.75 \pm 0.28$ | $0.76 \pm 0.01$ |
| S. sludge, 60 min | none | $63.4 \pm 1.2$ | $8.95 \pm 0.16$ | $2.55 \pm 0.36$ | $0.88 \pm 0.09$ |
| S. sludge, Standard | Zeolite | $71.07 \pm 0.44$ | $10.06 \pm 0.14$ | $2.82 \pm 0.11$ | $0.73 \pm 0.09$ |
| S. sludge, Standard | $CuSO_4$ 2 | $72.42 \pm 0.55$ | $10.55 \pm 0.34$ | $2.42 \pm 0.07$ | $0.29 \pm 0.02$ |
| S. sludge, Standard | $ZnSO_4$ 2 | $73.75 \pm 0.39$ | $10.71 \pm 0.33$ | $1.8 \pm 0.23$ | $0.44 \pm 0.06$ |
| S. sludge, Standard | $H_7[P(Mo_2O_7)_6]$ | $73.61 \pm 0.79$ | $11.01 \pm 0.35$ | $1.81 \pm 0.07$ | $0.50 \pm 0.06$ |
| S. sludge, Standard | $V_2O_5$ | $72.29 \pm 0.42$ | $10.39 \pm 0.13$ | $2.29 \pm 0.17$ | $0.68 \pm 0.04$ |
| S. sludge, Standard | MgO | $62 \pm 0.44$ | $8.74 \pm 0.17$ | $2.42 \pm 0.15$ | $0.73 \pm 0.02$ |
| S. sludge, Standard | $Al_2O_3$ | $69.37 \pm 0.28$ | $9.57 \pm 0.10$ | $2.69 \pm 0.05$ | $0.73 \pm 0.01$ |
| S. sludge, Standard | $CoCl_6$ | $72.37 \pm 0.84$ | $10.49 \pm 0.42$ | $1.95 \pm 0.31$ | $0.33 \pm 0.05$ |
| S. sludge, Standard | $NH_4Fe(SO_4)_2 \cdot 12H_2O$ | $72.56 \pm 0.41$ | $10.69 \pm 0.41$ | $2.37 \pm 0.45$ | $0.69 \pm 0.13$ |
| S. sludge, Standard | $NaHCO_3$ | $71.84 \pm 1.08$ | $9.71 \pm 0.25$ | $3.75 \pm 0.06$ | $1.02 \pm 0.04$ |
| S. sludge, Standard | KOH | $71.06 \pm 0.61$ | $9.69 \pm 0.22$ | $3.31 \pm 0.12$ | $1.15 \pm 0.04$ |
| S. sludge, Standard | $MoO_3$ | $72.26 \pm 0.78$ | $10.21 \pm 0.12$ | $2.40 \pm 0.06$ | $0.60 \pm 0.01$ |
| S. sludge, Standard | $NiSO_4$ | $72.66 \pm 0.59$ | $10.48 \pm 0.07$ | $1.85 \pm 0.09$ | $0.29 \pm 0.03$ |
| S. sludge + P. sludge (1:1), Standard | none | $72.105 \pm 0.67$ | $10.77 \pm 0.18$ | $4.20 \pm 0.15$ | $0.95 \pm 0.02$ |
| S. sludge + P. sludge (1:1), Standard | $NH_4Fe(SO_4)_2 \cdot 12H_2O$ | $71.46 \pm 0.24$ | $10.14 \pm 0.17$ | $3.95 \pm 0.24$ | $0.94 \pm 0.07$ |
| *Chlorella*, Standard | none | $68.84 \pm 0.21$ | $9.09 \pm 0.36$ | $6.54 \pm 0.03$ | $0.89 \pm 0.07$ |
| *Ulva*, Standard | none | $70.38 \pm 0.16$ | $8.99 \pm 0.10$ | $6.76 \pm 0.03$ | $0.89 \pm 0.02$ |
| *Polysiphonia*, Standard | none | $64.93 \pm 1.32$ | $8.55 \pm 0.83$ | $6.73 \pm 0.21$ | $1.22 \pm 0.12$ |
| Soy husk, 280 grad. | none | $67.36 \pm 0.4$ | $8.02 \pm 0.07$ | $8.59 \pm 0.09$ | $0.57 \pm 0.03$ |
| *Phragmites vulgaris* | none | $66.71 \pm 0.6$ | $7.16 \pm 0.32$ | $4.24 \pm 0.15$ | $0.43 \pm 0.03$ |
| *Typha angustifolia* L. | none | $65.26 \pm 0.22$ | $7.37 \pm 0.06$ | $4.09 \pm 0.04$ | $0.40 \pm 0.01$ |
| *Schoenoplectus lacustris* | none | $66.73 \pm 0.09$ | $7.54 \pm 0.10$ | $4.09 \pm 0.01$ | $0.60 \pm 0.02$ |
| BWW, Standard | none | $63.64 \pm 1.12$ | $7.29 \pm 0.21$ | $0.62 \pm 0.01$ | $0.35 \pm 0.04$ |
| BWW, Standard | $NH_4Fe(SO_4)_2 \cdot 12H_2O$ | $65.6 \pm 0.32$ | $7.09 \pm 0.02$ | $0.47 \pm 0.15$ | $0.01 \pm 0.01$ |
| BWW, Standard | $Al_2O_3$ | $64.56 \pm 6.22$ | $7.02 \pm 0.77$ | $0.46 \pm 0.07$ | $0.06 \pm 0.01$ |
| BWW, Standard | $KOH + NaHCO_3$ | $70.77 \pm 0.42$ | $7.90 \pm 0.10$ | $0.65 \pm 0.08$ | $0.32 \pm 0.01$ |

[1]—Standard conditions: 20 min, 260 °C, 4.6 MPa, biomass: water ratio = 1:10.

Analysis of changes in nitrogen and sulfur content shows that secondary sludge and BWW have the lowest nitrogen content. This is obviously related to the initial biomass composition. The organic matter of BWW consists of 90–95% cellulose and polysaccharides [38], which contain virtually no nitrogen-rich proteins. Likewise, secondary sludge also contains only 10–20% proteins [21]. The high nitrogen content in bio-oil produced from microalgae (*Chlorella*) and macroalgae (*Ulva, Polysiphonia*), as well as from soybean production waste, is due precisely to the fact that these biomasses are rich in protein. The

presence of a significant amount of sulfated polysaccharides (alginates, karaginans, etc.) in the macroalgae composition tends to generate bio-oil with a significant sulfur content.

We see that the maximum sulfur content in the fuel is observed during processing of sludge from primary clarifiers, which is quite natural and is associated with the presence of sulfates and sulfate-containing organic compounds in the composition of the municipal wastewater being treated. This fact is also noted in the work of Zacher et al. [39] and should be taken into account when developing processes of reform of bio-oil from primary sludge because it is possible to contaminate catalytic systems. The lowest sulfur and nitrogen content in bio-oil, and, accordingly, the best fuel quality, is reached via HTL processing of BWW and secondary sludge. It should be noted that BWW processing provides a bio-oil yield of 7–15%, which is two to four times less than the yield with sludge processing. The low nitrogen content in fuel during processing of lignin-containing biomass of terrestrial plants also fits into the general picture obtained by a number of previous studies [6,40] (Figure 4).

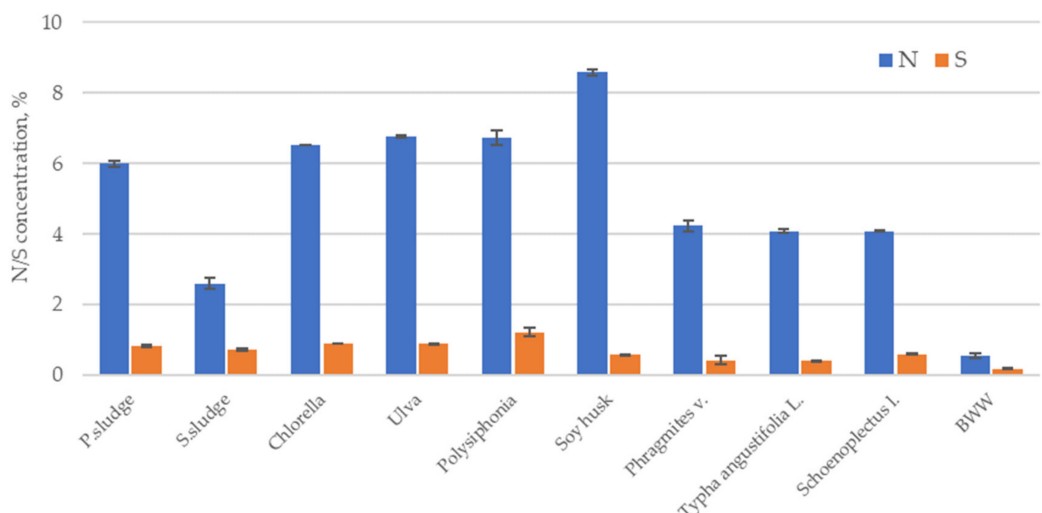

**Figure 4.** Concentration of N and S in bio-oil depending on raw material.

According to Tissot [41], the degree of kerogen maturity and the capillary oil quality are determined based on the H/C ratio. The higher the H/C ratio, the higher the oil quality, the larger the ratio of saturated alkanes and the smaller the ratio of oxygen-containing substances. That is why this parameter was used for bio-oil quality estimation. The H/C ratio for different types of biomass is shown in Figure 5. Oil produced from secondary sludge has the highest quality. When it comes to primary sludge, bio-oil produced from macro- and microalgae does not differ much in quality. A low degree of saturation and a correspondingly high proportion of oxygen-containing compounds is typical for biofuels produced from BWW, different types of aquatic plants (reeds, cattails, etc.) and soybean production waste. This is because breakdown of polysaccharides (the main components of this type of biomass) during hydrothermal conversion causes formation of alcohols, acids and aldehydes. In contrast, formation of alkanes, alkenes and aromatic hydrocarbons (the high content of which increases the H/C ratio) occurs during conversion of fats, as well as proteins to a lesser extent. The degree of saturation of oil obtained from lignin-based feedstock was never very high, which parallels the data obtained by others (Halleraker et al.) [7].

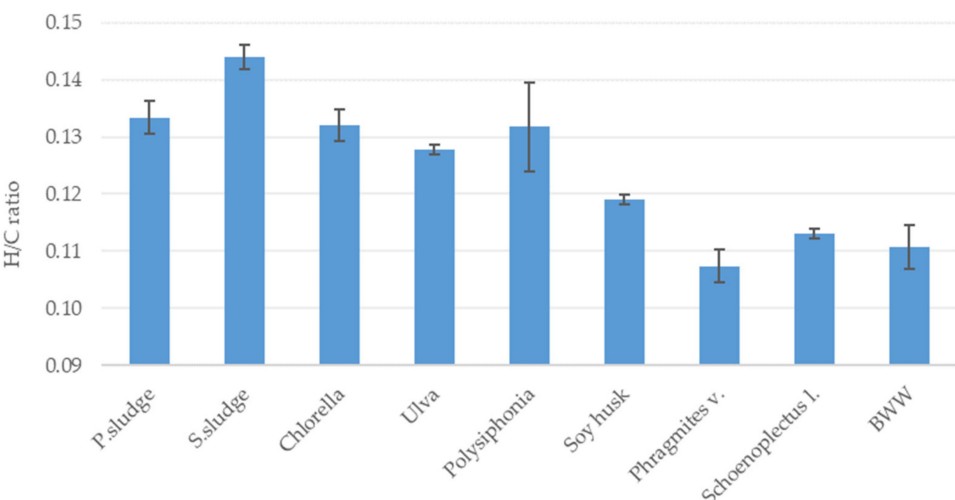

**Figure 5.** Changes in the H/C ratio of bio-oil depending on raw material.

A noteworthy result was obtained when comparing the H/C ratio for bio-oil produced from secondary sludge when using different process durations (Figure 6).

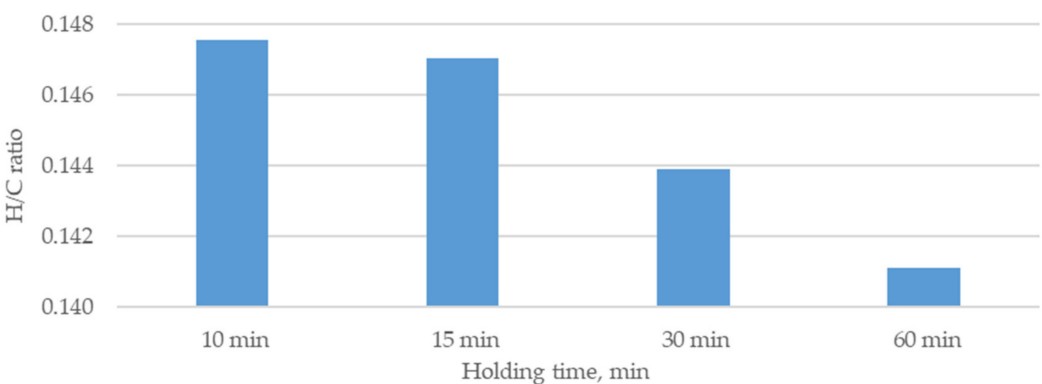

**Figure 6.** Changes in H/C ratio of bio-oil depending on holding time. Raw material—secondary sludge. Process conditions: 20 min, 260 °C, 4.6 MPa, biomass: water ratio = 1:10.

It was found that the saturation degree of bio-oil decreases significantly with increasing process time. This phenomenon is most likely associated with partial recombination and hydrolysis of carbohydrates generated at the first stage with formation of aromatic alcohols, acids and aldehydes. If there are significant levels of nitrogen, formation of condensed heterocyclic compounds (which are also characterized by lower hydrogen content) is possible. Thus, it obviously becomes a necessity to find the optimal duration of the process for each temperature and type of biomass. For secondary sludge, 10–20 min should be considered the optimal time for the process because it allows for maximum fuel output and produces the highest-quality fuel.

*3.3. Component Composition of Bio-Oils*

Analysis of a complex heterogeneous system of organic compounds with NMR spectroscopy enables researchers to glean a generalized impression of the types of compounds that are present. At the same time, the most complete picture can be obtained by combining carbon and hydrogen analyses. The results of NMR spectroscopic analysis for oils produced from various raw materials are presented in Tables 4 and 5, as well as in Figures 7 and 8.

**Table 4.** Content of the main groups of substances (in mass %) in bio-oil produced from various types of raw materials (based on NMR spectroscopy, 1H analysis).

| Type of Raw Material, Conditions | COOH, CHO, ArOH (8.2–12.0) [1] | R-OH, -CH2-O-R, Alkenes (4.2–6.0) | R-CH2-O-R, CH3-O-R (3.0–4.2) | R-CH2-CH=O (2.0–3.0) | Aromatic Hydrocarbons, Alkynes (6.0–8.2) | Aliphatic-H (0–2.0) |
|---|---|---|---|---|---|---|
| S. sludge, standard | 0.26 | 6.67 | 6.63 | 10.05 | 3.26 | 73.13 |
| P. sludge, standard | 0.165 | 2.725 | 5.775 | 10.955 | 8.01 | 72.37 |
| P. + S. sludge, standard | 0.26 | 3.53 | 8.86 | 14.15 | 11.45 | 61.75 |
| Furcellaria, standard | 0.59 | 9 | 10.19 | 30.63 | 14.68 | 34.91 |
| *Ulva* | 0.44 | 7.48 | 9.11 | 16.19 | 9.8 | 56.98 |
| *Chlorella v.* | 0.09 | 6.85 | 4.4 | 11.43 | 9.48 | 67.75 |
| *Schoenoplectus lac.* | 1.14 | 3.83 | 7.37 | 19.33 | 18.37 | 49.96 |
| *Phragmites v.* | 2.17 | 8.28 | 8.08 | 27.72 | 12.67 | 40.99 |
| *Typha ang.* | 2.27 | 9.82 | 10.82 | 30.46 | 11.3 | 35.33 |
| BWW | 0.7 | 13.22 | 14.2 | 27.97 | 15.5 | 28.41 |
| Soybean waste | 1.49 | 2.27 | 6.72 | 20.42 | 12.61 | 56.49 |

[1]—Chemical shift ranges (ppm).

**Table 5.** Content of the main groups of substances (in mass %) in bio-oil produced from different types of raw materials (based on NMR spectroscopy, 13C analysis).

| Type of Raw Material, Conditions | Aldehydes, Ketones (220–180) [1] | Acids and Derivatives (180–160) | Alcohols, Esters, Sugars (160–105) | Pure Aromatic, No Substitution (140–125) | Aromatic Hydrocarbons (160–105) | CH$_3$O Group in Lignin (60–55) | Aliphatic Carbohydrates (55–1) |
|---|---|---|---|---|---|---|---|
| S. sludge, standard | 0 | 3.34 | 0.64 | 4.06 | 4.06 | 0.51 | 91.45 |
| P. sludge, standard | 0 | 4.17 | 0.69 | 6.04 | 13.71 | 2.85 | 78.60 |
| P. + S. sludge, standard | 0 | 5.75 | 1.18 | 8.46 | 12.74 | 5.20 | 75.13 |
| *Furcellaria*, standard | 0.69 | 9.33 | 0.02 | 13.05 | 16.06 | 3.85 | 70.05 |
| *Ulva* | 0.22 | 4.94 | 0.17 | 12.04 | 12.54 | 0,83 | 81.30 |
| *Chlorella v.* | 0.05 | 4.48 | 0.08 | 13.06 | 13.14 | 2.01 | 80.24 |
| *Schoenoplectus lac.* | 2.43 | 2.26 | 2.14 | 10.49 | 23.41 | 2.32 | 67.44 |
| *Phragmites v.* | 4.47 | 10.33 | 2.35 | 10.96 | 22.79 | 6.26 | 53.80 |
| *Typha ang.* | 3.75 | 5.19 | 1.81 | 5.67 | 21.68 | 2.66 | 64.91 |
| BWW | 5.23 | 3.60 | 1.81 | 17.53 | 42.66 | 6.12 | 40.58 |
| Soybean waste | 0.41 | 6.54 | 3.85 | 15.91 | 22.43 | 1.91 | 64.86 |

[1]—Chemical shift ranges (ppm).

Based on NMR spectroscopy data, it can be concluded that aliphatic hydrocarbons and their derivatives predominate in bio-oil composition. The ratio of aliphatic carbon in bio-oils ranges from 40.6% to 91.5%. The maximum aliphatic carbon content is observed in bio-oil produced from secondary sludge (91.5%), followed by primary sludge (78.6%) and their mixture (75.13%). As expected, the maximum total content of aromatic carbon without substitutions (17.53%) is observed during processing of BWW and reed biomass (18.37%) due to the presence of lignin. This is also associated with a fairly high content of aromatic carbon in the oil produced from reed.

A surprising fact is that oil from macroalgae *Furcellaria* and *Chlorella* also has a high content of aromatic carbon and hydrogen. Earlier studies determined [26] that algae *Furcellaria* are characterized by a high content of proteins (25.4%) and fats (4.3%). As is well-established, the breakdown of long-chain fatty acids and subsequent cyclization of carbon chain leads to formation of aromatic and polyaromatic compounds [22]. Transformation of amino acids in turn can lead to formation of nitrogen-containing aromatic and polyaromatic compounds of the pyridine type. Thus, despite apparent strangeness, the presence of a significant amount of aromatic carbon in the composition of oil produced from macroalgae *Furcellaria* and microalgae *Chlorella* is in fact reasonable.

The high content of aromatic carbon in the composition of bio-oil from soybean production waste is explained by the presence of a significant amount of fats (10–12%) and proteins (53–55%) in the input biomass [42].

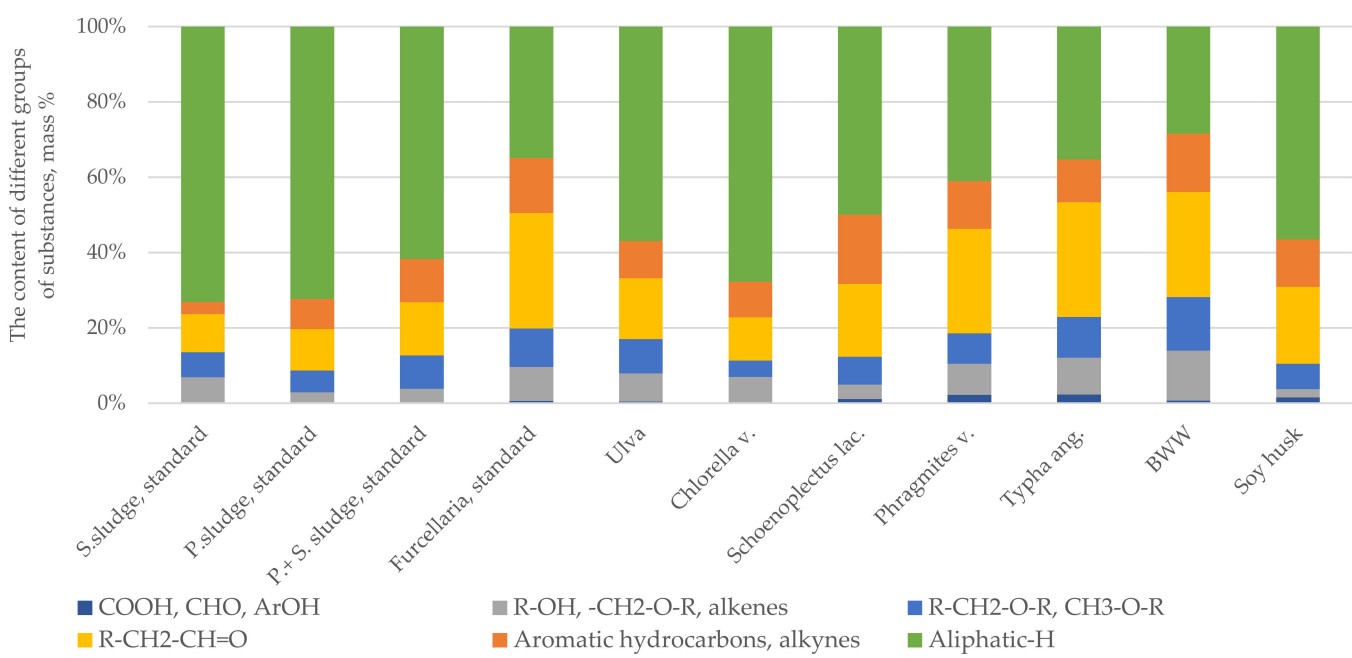

**Figure 7.** Content of the main groups of substances in bio-oil produced from various types of raw materials (based on NMR spectroscopy, 1H analysis).

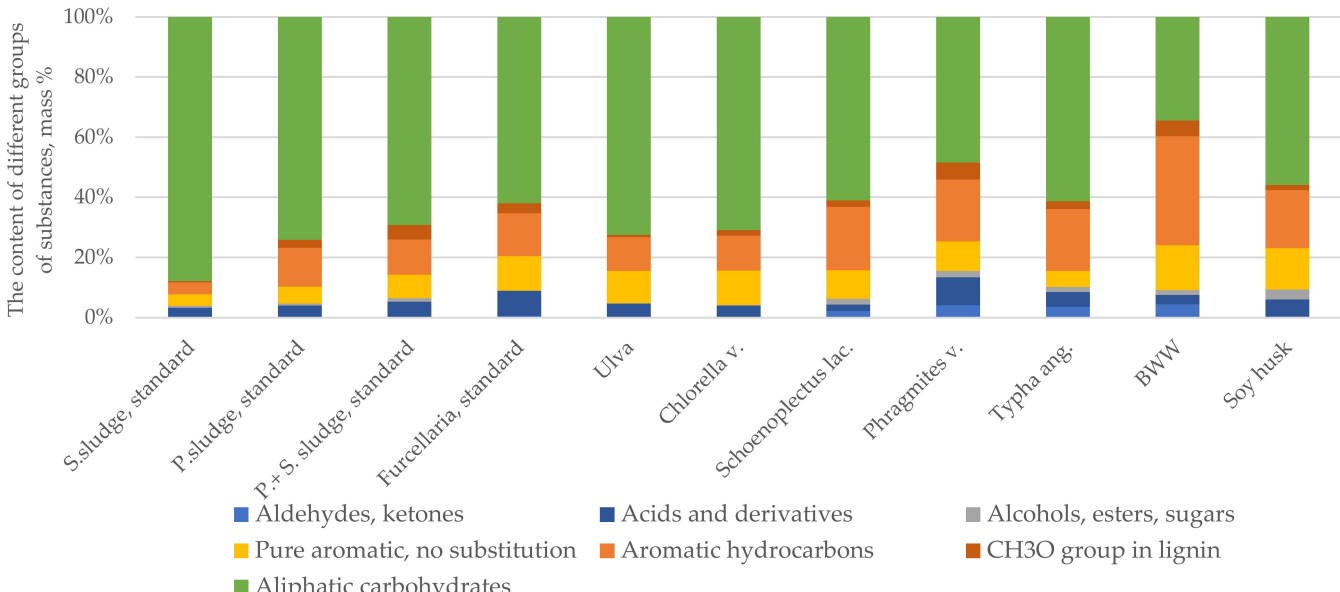

**Figure 8.** Content of the main groups of substances in bio-oil produced from different types of raw materials (based on NMR spectroscopy, 13C analysis).

Based on analysis of NMR spectra, it can also be noted that aromatic compounds are about half represented by derivatives with one or more oxygen-containing groups in their composition. This is also true for aromatic hydrocarbons formed during hydrolysis of lignin to form coniferyl, paracoumaryl and sinapyl alcohols [5].

The number of hydrogen and carbon ions bound to the oxygen atom is moderately high, with ranges of 19.6–56.1% (1H) and 4–12.5% (13C). The number of oxygen bonds is two or more times lower in bio-oil produced from primary and secondary sludge than it is in oil produced from plant biomass and BWW.

The results of NMR spectroscopy made it possible to choose a reliable method of GC–MS analysis. Since bio-oil is a multicomponent mixture, identification of the entire

variety of components is not possible and does not make sense in practice. With this in mind, it was decided to analyze the first ten peaks of the predominant substances.

The results of chromatographic analysis are presented in Table 6. The calculations were carried out as follows: the maximum peak area was taken as 100%; the remaining peaks were calculated as a ratio of the maximum peak. An example of a chromatogram is shown in Figure 9.

**Table 6.** Content of predominant substances in composition of bio-oil produced from various types of raw materials (based on GC–MS analysis).

| Substance | Area % | Proportion of Substances, % of the Mass of 10 Pre-Existing Components |
|---|---|---|
| *Bio-oil from secondary sludge* | | |
| n-Hexadecanoic acid | 100 | 56.1 |
| Hexadecanoic acid, methyl ester | 24.68 | 13.8 |
| Tetradecanoic acid | 12.43 | 7.0 |
| Hexadecanamide | 8.15 | 4.6 |
| Octadecanoic acid | 7.43 | 4.2 |
| 9-Octadecenoic acid (Z)-, methyl ester | 7.32 | 4.1 |
| Methyl stearate | 6.98 | 3.9 |
| 12-Octadecenoic acid, methyl ester | 4.58 | 2.6 |
| Methyl tetradecanoate | 3.51 | 2.0 |
| trans-Vaccenic acid (110-jctadecenoic acid) | 3.31 | 1.9 |
| *Bio-oil from primary sludge* | | |
| n-Hexadecanoic acid | 100 | 42.0 |
| Cyclo-(L-leucyl-L-phenylalanyl) | 52.54 | 22.0 |
| Hexadecanamide | 36.3 | 15.2 |
| Cyclo(L-prolyl-L-valine) | 33.07 | 13.9 |
| 3,6-Diisopropylpiperazin-2,5-dione | 26.58 | 11.2 |
| Tetradecanoic acid | 24.63 | 10.3 |
| Palmitoleic acid | 20.58 | 8.6 |
| Pyrrolo [1,2-a]pyrazine-1,4-dione, hexahydro-3-(2-methylpropyl)- | 16.28 | 6.8 |
| Oleic Acid | 14.36 | 6.0 |
| 2,5-Piperazinedione, 3-methyl-6-(1-methylethyl)- | 13.98 | 5.9 |
| *Bio-oil from BWW* | | |
| Vanilin | 100 | 30.9 |
| 2-Cyclopentet-1-one, 2-hydroxy-3-methyl | 63.97 | 19.8 |
| 2-Propanone, 1-(4-hydroxy-3-methoxyphenyl)- | 47.47 | 14.7 |
| Tetradecanoic acid | 22.04 | 6.8 |
| Phenol, 2-methoxy | 20.9 | 6.5 |
| n-Hexadecanoic acid | 19.83 | 6.1 |
| 3,4-Divanillyltetrahydrofuran | 16.21 | 5.0 |
| Phenol, 2-methoxy-5-(1-propenyl)-, (E)- | 12.4 | 3.8 |
| Apocynin | 11.21 | 3.5 |
| Phenol, 4-ethyl-2-methoxy- | 9.62 | 3.0 |
| *Bio-oil from algae Furcellaria* | | |
| 2,5-Piperazinedione, 3,6-bis(2-methylpropyl)- | 100 | 37.7 |
| Cyclo-(L-leucyl-L-phenylalanyl) | 39.04 | 14.7 |
| n-Hexadecanoic acid | 29.9 | 11.3 |
| 2,5-Piperazinedione, 3-benzyl-6-isopropyl- | 17.72 | 6.7 |
| Pyrrolo [1,2-a]pyrazine-1,4-dione, hexahydro-3-(phenylmethyl)- | 15.88 | 6.0 |
| Pyrrolo [1,2-a]pyrazine-1,4-dione, hexahydro-3-(2-methylpropyl)- | 14.98 | 5.6 |
| 3-Acetyl-9-methylcarbazole | 14.59 | 5.5 |
| 6-Octadecenoic acid | 13.34 | 5.0 |
| Cyclo(L-prolyl-L-valine) | 12.06 | 4.5 |
| Tetradecanoic acid | 8.05 | 3.0 |

Analysis of the main bio-oil substances from activated sludge shows that bio-oil is mainly represented by fatty acids (95% of the total area of the ten predominant substances). Hexadecanamide, a fatty amide, which is a carboxamide derived from palmitic acid, is also included in the ten predominant substances of bio-oil produced from sludge.

The oil produced from primary sludge differs in composition. Fatty acid (n-Hexadecanoic acid) has the largest peak area, and the total fatty acid content is 67.0%. A significant portion (up to 23%) of the bio-oil produced from primary sludge falls within the category of nitrogen/oxygen-containing compounds (piperazinedione, amides, etc.). This is associated with a higher protein content in primary sludge composition.

Vanillin, which belongs to hydroxy-benzaldehydes, is the predominant component in oils produced from BWW. Such a finding is quite predictable because BWW biomass is rich in lignin, which has been used for synthesis of guaiacol and vanillin [43]. This fact once again points us towards the practical idea of organizing a sequence of hydrothermal conversion involving release of valuable products from organic waste biomass at the first stage, followed by their subsequent processing into liquid fuel. The remaining predominant substances of bio-oil from BWW are represented by phenolic alcohols (62% of the mass of the predominant substances), cyclic oxygen-containing carbon–hydrogens of the limiting range (19.8%) and fatty acids (12.9%). This fact reaffirms the expediency of the aforementioned idea to organize a sequence of hydrothermal biomass conversion into liquid fuel.

In the oil produced from microalgae *Furcellaria*, the share of acid/nitrogen-containing compounds (piperazinedione, amides, methyl carbazole derivatives, etc.) accounts for more than 75%, and the share of fatty acids is 19.3%.

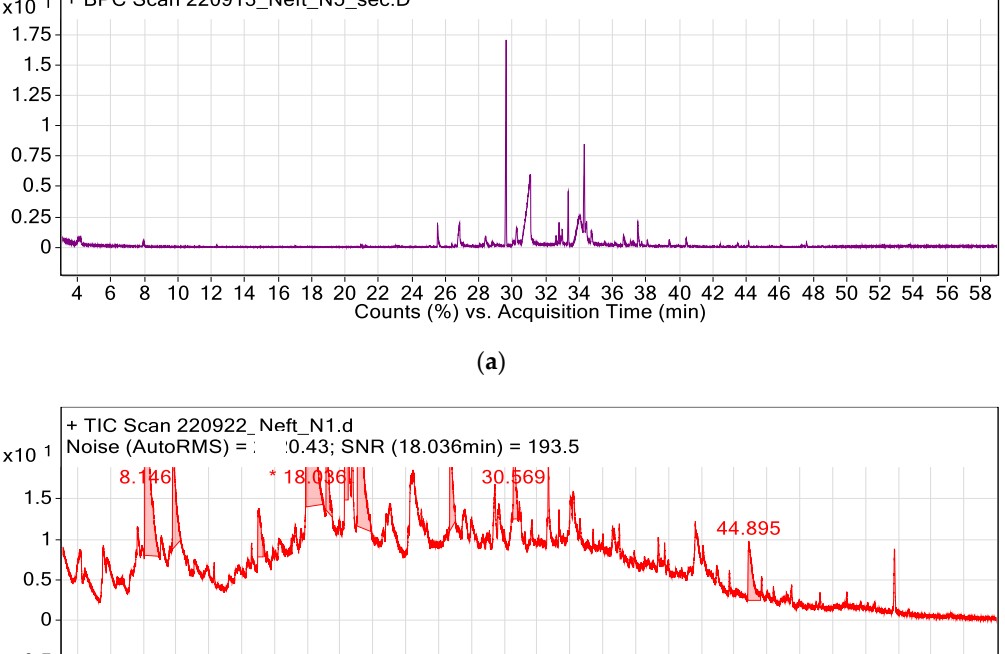

**Figure 9.** Example of a bio-oil chromatogram: (**a**) raw material: secondary sludge, 260 °C, without catalysts; (**b**) raw material: BWW, 260 °C, without catalysts.

## 4. Conclusions

Analysis of the bio-oil fractional composition by boiling points showed that the highest levels of components with a boiling point of up to 350 °C (gasoline and diesel-like)

during conversion of biomass, without addition of catalysts, are reached while processing secondary sludge (58.5%) and primary sludge (59.9%).

The positive effect of catalytic systems during sewage sludge conversion was demonstrated not only with regard to yield volume but also in terms of bio-oil composition. In fact, when using almost all catalysts, an increase in the yield of light-burning oil fractions was noted. However, the best results in improving fuel quality were achieved when using copper, iron and nickel salts as catalysts, which increased the share of fraction with a boiling point below 350 °C by 18.8%, 22.2% and 24.5%, respectively.

Analyzing the elemental composition of bio-oils allowed us to establish that the highest ratios of H/C (as an indicator of oil quality) were achieved with conversion of secondary sludge (0.161) and primary sludge (0.150). The highest levels of interfering elements were observed in bio-oil produced from soybean production waste (8.4–8.9%), primary sludge (6.8–7.5%), macroalgae *Ulva* (6.8%) and *Polysiphonia* (6.7%).

Based on NMR spectroscopy data, it was found that aliphatic carbon prevails (40.6–91.5%) in bio-oil composition. The highest levels of aliphatic carbon are observed in bio-oil produced from secondary sludge (91.5%), primary sludge (78.6%) and their mixture (75.1%). The maximum total content of aromatic carbon without substitutions (17.5%) is observed during processing of BWW and with reed biomass (18.4%), which is due to the presence of a significant proportion of lignin. The share of hydrogen and carbon ions bound to the oxygen atom varies in all types of oil but generally falls between 19.6 and 56.1% (1P) and 4 and 12.5% (13C). At the same time, the number of oxygen bonds is two or more times lower in bio-oils produced from primary and secondary sludge than in fuels produced from plant biomass and BWW.

These results were confirmed by GC–MS studies. Thus, the main components of bio-oil from primary and secondary sludge are fatty acids, but a significant part (up to 23%) of bio-oil produced from sludge falls into the category of nitrogen/oxygen-containing compounds of the piperazinedione type and amides.

This study identified that the oil produced from secondary sludge has the best quality, characterized by a high content of low-weight aliphatic compounds (with a boiling point of up to 350 °C), along with insignificant levels of nitrogen, sulfur and oxygen. Given the significant volumes of secondary sludge generated in almost all settlements that have a biological wastewater treatment plant, this type of waste should be considered as the most promising avenue towards development of technology for producing liquid fuel by hydrothermal liquefaction. For more complete use of carbohydrates contained in the sludge and implementation of the Maillard reaction, it is recommended to use biomass with a high protein content (soybean husk or some types of microalgae) as an additional raw material that provides a synergistic effect.

**Author Contributions:** Conceptualization, O.B.; Data curation, Y.K.; Investigation, M.K. and N.O.; Project administration, O.B.; Resources, O.B.; Validation, N.S.; Writing—Original draft, Y.K., O.B. and M.K.; Writing—Review and editing, O.B., Y.K. and N.S. All authors have read and agreed to the published version of the manuscript.

**Funding:** The study was carried out with the financial support of the Ministry of Science and Higher Education of the Russian Federation, project number FZWM-2021–0016.

**Informed Consent Statement:** Not applicable.

**Data Availability Statement:** Not applicable.

**Conflicts of Interest:** The authors declare no conflict of interest.

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
