# Peer review of "Analysis and Comparison of Bio-Oils Obtained by Hydrothermal Liquefaction of Organic Waste"

_sustainability, doi:10.3390/su15020980_

Round 1
Reviewer 1 Report
The manuscript titled "Bio-Oils obtained by hydrothermal liquefaction waste: Analysis and comparison" presents new results about the chemical composition of different bio-oil obtained from different biomass/waste sources. The information is novel and quite interesting. Nevertheless, there are some major issues to be addressed before manuscript publication.
1. Please, revise English since there are wordy sentences in the manuscript. Excessive use of the word "carried out" please introduce some synonyms. Please, reduce or avoid the use of pronouns such as "us" in the manuscript. An English revision is recommended.
2. Introduction should be improved when describing the advantages of the HTL process and problem statement. The novelty of the word was stated clearly.
3. Materials and Methods. A brief global description of the raw material to be used in the study should be done to contextualize the reader about what to expect in the methodology section.
4. Line 66. Please, introduce geographical coordinates.
5. Line 84. Please, introduce more information about the experiments (operating conditions) without mattering previous reports. The authors can cite previous research to complement the given information in the manuscript but essential data should be provided (e.g., temperature, solid-to-liquid ratio, residence time, catalyst use, etc).
6. Line 143 - 144. Where was mentioned the use of a catalyst to perform the HTL process? Please, improve the materials and methods section to involve all this information.
7. Line 162. What is the minimum residence time proposed by the authors to carry out the HTL process for each raw material since longer residence times do not have a significant effect?
8. Results and discussion. The authors made an analysis of the experimental data. Nevertheless, discussion and comparison with other reports in the open literature are scarce. Please, made a more in-depth analysis comparing the results with the open literature (at least regarding yields, and other characteristics such as H/C ratio, etc).
9. Please, at the end of the manuscript introduce a list of the best to the worst raw material evaluated to produce bio-oil based on yields, elemental analysis, and chemical composition. What will be the potential interactions between the biomass/waste source and a possible HTL plant?
10. Conclusions. Please, reduce the conclusions. The authors should be more concise.
Author Response
Dear reviewer,
many thanks for the work done. Thanks to your comments, our article has become more logical and structured. I ask you to read the answers to your comments in the attached file.
Sincerely, the team of authors.

Reviewer 2 Report
The paper presented a comprehensive study examining biofuel composition during the process of catalytic and non-catalytic conversion of sewage sludge, sludge from food processing plants, and BWW. In this work, composition analysis of the bio-oils was carried out using
chromatography and nuclear magnetic resonance spectroscopy.
The manuscript is well-organized beginning by describing the Materials and methods in details. Then the results and conclusion are explained well.
Only one inquiry, in line 75, " Before processing, all samples were dried at 60°C, ". the authors should explain why choosing 60°C.
Author Response
Thank you for reviewing the article and good feedback. You have only one inquiry: “the authors should explain why choosing 60°C for samples drying”. We added the following explanation:
“The drying temperature 60°C was chosen because some authors proved that at higher temperatures composition of the biomass occurring, in particular, the partial destruction of proteins and lignin [Nunes, L.J. R.; Matias, J. C. O.; Catalão, J. P. S. Chapter 2 - Physical Pretreatment of Biomass. In Torrefaction of Biomass for Energy Applications; Publisher: Academic Press: Cembridge, 2018; pp. 45-88. https://doi.org/10.1016/B978-0-12-809462-4.00002-X.].
Reviewer 3 Report
The subject matter of the article is well known, especially for biomass waste. Nevertheless, it should be emphasized that the authors present the results of extensive experimental research that may be of interest to researchers.
I believe that the article should be published after taking into account the comments:
The literature review is very modest and needs supplementing. Authors should indicate what is new in their article.
- line 218: Table 3: the composition is less than 100%, what is the rest?
Author Response
Thank you for reviewing the article and good feedback. You have only 2 comments
1.The literature review is very modest and needs supplementing. Authors should indicate what is new in their article.
The introduction has been updated and expanded with a special focus on novelty.
- Line 218: Table 3: the composition is less than 100%, what is the rest?
The rest is mostly oxygen. Obviously it could be over elements, but in trace concentration.
Best regards
Round 2
Reviewer 1 Report
The authors have made the suggested changes. The manuscript can be accepted for publication.